# ON THE EFFECTIVENESS OF
# WEIGHT-ENCODED NEURAL IMPLICIT 3D SHAPES

## ABSTRACT

A neural implicit outputs a number indicating whether the given query point in space is inside, outside, or on a surface. Many prior works have focused on *latent-encoded* neural implicits, where a latent vector encoding of a specific shape is also fed as input. While affording latent-space interpolation, this comes at the cost of reconstruction accuracy for any *single* shape. Training a specific network for each 3D shape, a *weight-encoded* neural implicit may forgo the latent vector and focus reconstruction accuracy on the details of a single shape. While previously considered as an intermediary representation for 3D scanning tasks or as a toy-problem leading up to latent-encoding tasks, weight-encoded neural implicits have not yet been taken seriously as a 3D shape representation. In this paper, we establish that weight-encoded neural implicits meet the criteria of a first-class 3D shape representation. We introduce a suite of technical contributions to improve reconstruction accuracy, convergence, and robustness when learning the signed distance field induced by a polygonal mesh — the *de facto* standard representation. Viewed as a lossy compression, our conversion outperforms standard techniques from geometry processing. Compared to previous latent- and weight-encoded neural implicits we demonstrate superior robustness, scalability, and performance.

## 1 INTRODUCTION

While 3D surface representation has been a foundational topic of study in the computer graphics community for over four decades, recent developments in machine learning have highlighted the potential that neural networks can play as effective parameterizations of solid shapes.

The success of neural approaches to shape representations has been evidenced both through their ability of representing complex geometries as well as their utility in end-to-end 3D shape learning, reconstruction, and understanding and tasks. These approaches also make use of the growing availability of user generated 3D content and high-fidelity 3D capture devices, e.g., point cloud scanners.

For these 3D tasks, one powerful configuration is to represent a 3D surface $\mathcal{S}$ as the set containing any point $\vec{x} \in \mathbb{R}^3$ for which an implicit function (i.e., a neural network) evaluates to zero:

$$\mathcal{S} := \big\{ \vec{x} \in \mathbb{R}^3 | f_\theta(\vec{x}; \vec{z}) = 0 \big\}, \tag{1}$$

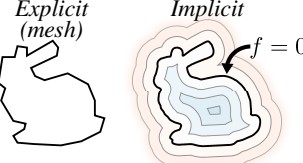

where $\theta \in \mathbb{R}^m$ are the network weights and $\vec{z} \in \mathbb{R}^k$ is an input latent vector encoding a particular shape. In contrast to the *de facto* standard polygonal mesh representation which *explicitly* discretizes a surface's geometry, the function $f$ *implicitly* defines the shape $\mathcal{S}$ encoded in $\vec{z}$. We refer to the representation in Eq. (1) as a *latent-encoded neural implicit*.

Park et al. (2019) propose to optimize the weights $\theta$ so each shape $\mathcal{S}_i \in \mathcal{D}$ in a dataset or shape distribution $\mathcal{D}$ is encoded into a corresponding latent vector $\vec{z}_i$. If successfully trained, the weights $\theta$ of their DEEPSDF implicit function $f_\theta$ can be said to generalize across the "shape space" of $\mathcal{D}$. As always with supervision, reducing the training set from $\mathcal{D}$ will affect $f$'s ability to generalize and can lead to overfitting. Doing so may seem, at first, to be an ill-fated and uninteresting idea.

Our work considers an extreme case – when the training set is reduced to a single shape $\mathcal{S}_i$. We can draw a simple but powerful conclusion: in this setting, one can completely forgo the latent vector

(i.e., $k = 0$). From the perspective of learning the shape space of $\mathcal{D}$, we can "purposefully overfit" a network to a single shape $\mathcal{S}_i$:

$$\mathcal{S}_i := \left\{ x \in \mathbb{R}^3 | f_{\theta_i}(x) = 0 \right\}, \tag{2}$$

where $\theta_i$ now parameterizes a *weight-encoded neural implicit* for the single shape $\mathcal{S}_i$.

In the pursuit of learning the "space of shapes," representing a single shape as a weight-encoded neural implicit has been discarded as a basic validation check or stepping stone toward the ultimate goal of generalizing over many shapes (see, e.g., (Chen & Zhang, 2019; Park et al., 2019; Atzmon & Lipman, 2020a;b)). Weight-encoded neural implicits, while not novel, ***have been overlooked*** as a valuable shape representation beyond learning and computer vision tasks. For example, the original DEEPSDF work briefly considered – and nearly immediately discards – the idea of independently encoding each shape of a large collection:

> "*Training a specific neural network for each shape is neither feasible nor very useful.*"
> – Park et al. (2019)

We propose training a specific neural network for each shape and will show that this approach is both feasible and very useful.

We establish that a weight-encoded neural implicit meets the criteria of a first-class representation for 3D shapes ready for direct use in graphics and geometry processing pipelines (see inset table) While common solid shape representations have some important features and miss others, neural implicits provide a new and rich constellation of features. Unstructured point clouds are often raw output from 3D scanners, but do not admit straightforward smooth surface visualization (I). While meshes are

| | I | II | III | IV |
|---|---|---|---|---|
| Point cloud | × | × | ● | ×/● |
| Mesh | ● | × | ● | × |
| Regular grid | ● | ● | × | ● |
| Adaptive grid | ● | ● | ● | × |
| Neural implicit | ● | ● | ● | ● |

the *de facto* standard representation, conducting signed distance queries and CSG operations remain non-trivial (II). Signed distances or occupancies stored on a regular grid admit fast spatial queries and are vectorizeable just like 2D images, but they wastefully sample space uniformly rather than compactly adapt their storage budget to a particular shape (III). Adaptive or sparse grids are more economical, but, just as meshes will have a different number of vertices and faces, adaptive grids will different storage profiles and access paths precluding consistent data vectorization (IV).

While previous methods have explored weight-encoded neural implicits as an *intermediary* representation for scene reconstruction (e.g., (Mildenhall et al., 2020)) and noisy point-cloud surfacing tasks (e.g., (Atzmon & Lipman, 2020a;b)), we consider neural implicits as the *primary* geometric representation. Beyond this observational contribution, our technical contributions include a proposed architecture and training regime for converting the (current) most widely-adopted 3D geometry format – polygonal meshes – into a weight-encoded neural implicit representation.

We report on experiments[1] with different architectures, sampling techniques, and activation functions – including positional encoding (Mildenhall et al., 2020) and sinusoidal activation approaches (Sitzmann et al., 2020b) that have proven powerful in the context of neural implicits. Compared to existing training regimes, we benefit from memory improvements (directly impacting visualization performance), stability to perturbed input data, and scalability to large datasets.

Weight-encoded neural implicits can be treated as an efficient, lossy compression for 3D shapes. Increasing the size of the network increases the 3D surface accuracy (see Figure 1) and, compared to standard graphics solutions for reducing complexity (mesh decimation and storing signed distances on a regular grid), we achieve higher accuracy for the same memory footprint as well as maintaining a SIMD representation: $n$ shapes can be represented as $n$ weight-vectors for a fixed architecture.

The benefits of converting an existing mesh to a neural implicit extends beyond compression: in approximating the signed distance field (SDF) of the model, neural implicits are both directly usable for many tasks in graphics and geometry processing, and preferable in many contexts compared to traditional representations. Many downstream uses of 3D shapes already mandate the conversion of meshes to less accurate grid-based SDFs, due to the ease and efficiency of computation for SDFs: here, neural implicits serve as a drop-in replacement.

---

[1] Source code, data, and demo at our (anonymized) repo: https://github.com/u2ni/ICLR2021

Many works explore latent-encoding methods (e.g., (Park et al., 2019; Atzmon & Lipman, 2020a;b)), taking advantage of interpolation in latent space as a (learned) proxy for exploration in the "space of shapes". We show that this flexibility comes at a direct cost of other desirable proprieties. In particular, we show that latent-

| Encoding: | Latent | Weight |
|---|---|---|
| Interpolation: | trivial | non-trivial |
| Scalability: | poor | excellent |
| Stability: | poor | excellent |

encoded neural implicits scale poorly as a representation for individual shapes both at training and inference time. Existing latent-encoded neural implicits are sensitive to the distribution of training data: while they may perform well for large datasets of a limited subclass of shapes (e.g., "jet airplanes"), we show that training fails with more general 3D shape datasets. Even within a class, existing methods rely on canonical orientation alignment (see Figure 2) in order to alleviate some of this difficulty – such orientation are notably (and notoriously) not present in 3D shapes captured or authored *in the wild* and, as a result, latent-encoded neural implicits will fail to provide meaningful results for many real-world and practical shape datasets. Fitting latent-encoded neural implicits to each shape independently complicates shape space interpolation, rendering it difficult though not impossible (Sitzmann et al., 2020a). In contrast, weight-encoded neural implicits leverage the power of the neural network function space without the constraints imposed by the requirement of generalizing across shapes through latent sampling.

## 2 METHOD

Neural implicits soared in popularity over the last year. While significant attention has been given to perfecting network architectures and loss functions in the context of latent-encoding and point-cloud reconstruction, there is relatively little consideration of the conversion process from 3D surface meshes to weight-encoded neural implicits (e.g., both Park et al. (2019) and Sitzmann et al. (2020b) consider this task briefly). We focus on identifying a setup to optimize weight-encoded neural implicits for arbitrary shapes robustly with a small number of parameters while achieving a high surface accuracy. Once successfully converted, we consider how the weight-encoded neural implicit representation compares to standard 3D model reduction techniques and how choosing this representation impacts downstream graphics and geometric modeling operations.

### 2.1 SIGNED DISTANCE FIELD REGRESSION

In general, the value of an implicit function $f$ away from its zero-isosurface can be arbitrary. In shape learning, many previous methods have considered occupancy where $f(\vec{x})$ outputs the likelihood of $\vec{x}$ being inside of a solid shape (and extract the surface as the 50%-isosurface) Mildenhall et al. (2020); Mescheder et al. (2019); Littwin & Wolf (2019); Chen & Zhang (2019); Maturana & Scherer (2015); Wang et al. (2018). We instead advocate that $f$ should approximate the *signed distance field* (SDF) induced by a given solid shape. Learning properties aside (see, e.g., (Park et al., 2019)), SDFs are more immediately useful in graphics and geometry processing applications.

Given a surface $\mathcal{S} = \partial\mathcal{V}$ of a volumetric solid $\mathcal{V} \subset \mathbb{R}^3$, the signed distance field $g_{\mathcal{S}} : \mathbb{R}^3 \to \mathbb{R}$ induced by $\mathcal{S}$ is a continuous function of space that outputs the distance of a query point $\vec{x} \in \mathbb{R}^3$ modulated by $\pm 1$ depending on whether $\vec{x}$ is inside or outside of the solid:

$$g_{\mathcal{S}}(\vec{x}) = \text{sign}_{\mathcal{S}}(\vec{x}) \min_{\vec{p} \in \mathcal{S}} \|\vec{x} - \vec{p}\|, \quad \text{where} \quad \text{sign}_{\mathcal{S}}(\vec{x}) = \begin{cases} -1 & \text{if } \vec{x} \in \mathcal{V}, \\ 1 & \text{otherwise.} \end{cases} \quad (3)$$

Our goal is to regress a feed-forward network $f_\theta$ to approximate the SDF of a given surface $\mathcal{S}$:

$$f_\theta(\vec{x}) \approx g_{\mathcal{S}}(\vec{x}). \quad (4)$$

If successfully trained, the weights $\theta \in \mathbb{R}^m$ encode a neural implicit representing $\mathcal{S}$.

#### 2.1.1 ARCHITECTURE

Our proposed architecture is a feed-forward fully connected network with $N$ layers, of hidden size $H$. Each hidden layer has ReLU non-linearities, while the output layer is activated by $\tanh$.

Increasing the depth and width of this network will generally improve accuracy but at the cost of increasing the memory footprint and, for example, the time required to render the surface. The

Figure 1: We visualize the role that varying the number of network layers and hidden layer sizes: (left to right) average reconstruction error, memory footprint and first-frame render time (DeepSDF, other setups, and our defaults in red, gray, and blue, respectively).

weight-encoded neural implicit's rendered in Figures 2, 4, and 8 all share a common architecture of just 8 fully connected layers with a hidden size of 32 (resulting in just 7553 weights, or 59 kB in memory). Through experimentation on a subset of 1000 meshes from Thingi10k (Zhou & Jacobson, 2016), we find that this configuration yields a good balance between reconstruction accuracy, rendering speed, and memory impact (Figure 1). While maintaining acceptable surface quality, our default architecture has a 99% reduction in number of parameters and 93% speed up in "time to render first frame" compared to the default weight-encoding architecture of (Park et al., 2019).

Excited by the recent work exploring methods to overcome an MLP's bias to learn low frequency signals faster, we performed experiments using both positional encodings (Tancik et al., 2020) and SIREN activations (Sitzmann et al., 2020b). Both perform well when the network architecture is sufficiently wide (e.g., $H > 64$), but introduce surface noise with our more compact architecture. See Appendix A.3 for detailed experimental setup and findings.

By increasing $N$ and $H$, our network could *in theory* (Hornik et al., 1989) learn to emulate any arbitrary topology shape with infinite precision. In reality, like any representation, there are trade-offs. The network complexity can be increased over our base configuration for smaller surface reconstruction error, or decreased for faster rendering speeds depending on the application. A sample of geometries produced at a number of configurations can be seen in Figure 1.

### 2.1.2 Integrated Loss → Importance Sampling

Particularly choices of pointwise loss functions have been well explored by previous papers (Park et al., 2019; Atzmon & Lipman, 2020a;b; Gropp et al., 2020; Sitzmann et al., 2020b), in our experiments we find that a simple absolute difference $|f_\theta(\vec{x}) - g_\mathcal{S}(\vec{x})|$ works well. Defining the total loss after the fact via *ad hoc* sampling (near-)surface sample process (Park et al., 2019; Atzmon & Lipman, 2020a;b) leaves an unclear notion whether the total loss can be expressed as an integral and hides possibly unwanted bias. We focus instead on how to integrate this pointwise loss over space.

Sampling based on mesh vertices (Littwin & Wolf, 2019; Sitzmann et al., 2020b) reduces accuracy in the middle of triangle edges and faces and introduces bias near regions of the mesh (inset: Vertex) with denser vertex distributions regardless of the geometric complexity or saliency of the region.

Similarly, sampling from Gaussians centered on the surface Park et al. (2019); Chen & Zhang (2019); Atzmon & Lipman (2020a;b) will place over emphasis in regions of high curvature, in thin solid/void regions (inset: Surface).

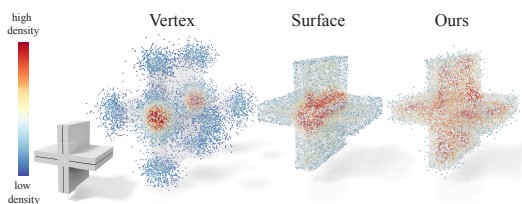

In contrast to *ad hoc* samplings, we define the total loss directly as an integral over space,

$$L(\theta) = \int_{\mathbb{R}^3} w(\vec{x}) \, |f_\theta(\vec{x}) - g_\mathcal{S}(\vec{x})| \; d\vec{x}, \qquad (5)$$

where $w : \mathbb{R}^3 \to \mathbb{R}_{\geq 0}$ is a non-negative weighting function with finite integral over $\mathbb{R}^3$.

Methods which randomly sample within a bounding box around a given shape (Mescheder et al., 2019; Tancik et al., 2020) can be understood as choosing $w$ to be the characteristic function of the box. As Park et al. (2019) already observe, this is wasteful if we care most that $f$ is accurate near the shape's surface (i.e., where $g_\mathcal{S} = 0$).

We achieve this *directly* — without yet invoking sampling — by choosing $w$ exponentially as distance to $\mathcal{S}$ grows, specifically:

$$w(\vec{x}) = e^{-\beta|g_{\mathcal{S}}(\vec{x})|}, \tag{6}$$

where $\beta \geq 0$ can be adjusted from uniform sampling ($\beta = 0$) to $\beta \to \infty$ for surface-only sampling.

Attempting to sample space and measure the integrand of Eq. (5) directly leads to many samples having little to no numerical effect during training. For example, if $\beta = 30$ and we consider a point unit distance away from the surface, the weighting term itself closes in on machine double precision $w \approx 9e-14$. By resisting the urge to prematurely sample until after we have written our total loss function as an integral, we can instead apply *importance sampling* (Kahn & Harris, 1951) to construct a proportional approximation:

$$L(\theta) \approx \sum_{\vec{x} \in \mathcal{D}_w} |f_\theta(\vec{x}) - g_{\mathcal{S}}(\vec{x})|, \tag{7}$$

where $\mathcal{D}_w$ is a distribution over $\mathbb{R}^3$ with probabilities proportional to $w$.

We sample from $\mathcal{D}_w$ in practice via a simple subset rejection strategy. Starting with a large (e.g., 10M) pool of uniform samples within a loose bounding sphere around the shape, we re-sample (with replacement) a smaller (e.g., 1M) subset with probability according to $w$. Further improvements may be possible by incorporating advanced sampling patterns *à la* Xu et al. (2020).

Compared to uniform sampling, weighting by our choice of $w$ leads to faster convergence and reduced surface reconstruction when validating against a subset of 1000 geometries from Thingi10k (96 epochs with surface error of 0.00231). Compared to the sampling of Park et al. (2019), we match convergence speed (86 epochs each) and demonstrate a $\approx 5\%$ improvement in surface error.

Perhaps the most valuable property of our importance sampling scheme to be its flexibility.

Our method has effectively removed all unintended bias present in previous approaches, and enables complete user control on **intended** bias to the sampling process. The importance metric, $w(\vec{x})$, can be modified to explicitly bias im-

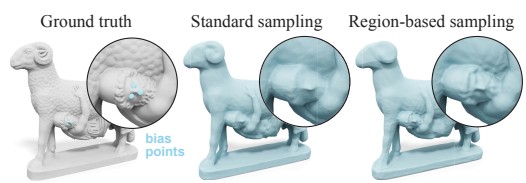

Ground truth    Standard sampling    Region-based sampling

bias points

portance toward regions of high curvature, minimum feature size (emulating the hidden bias of Park et al. (2019)), or near user annotations (see inset where $w(\vec{x})$ is additionally scaled according to user selection). This flexibility allows for greater use of the network's capacity on areas important to the user, without increasing overall network complexity or radically changing the sampling protocol.

## 2.2 ROBUST LOSS FUNCTION FOR MESHES IN THE WILD

The input $\mathcal{S}$ should be the boundary of a solid region $\mathcal{V} \subset \mathbb{R}^3$; that is, a closed, consistently oriented, non-self-intersecting surface. Ignoring "two-sided" meshes that are not intended to represent the boundary of a solid shape (e.g., clothing), many if not most meshes found online *which intend* to represent a solid shape would not qualify these strict pre-conditions. Zhou & Jacobson (2016)

observe that nearly 50% of Thingi10k's solid models for 3D printing fail one criteria or another. The failure point in terms of our equations so far is the definition of the signing function $\text{sign}(\vec{x})$ in Eq. (3) which relies on determining whether a point $\vec{x}$ lies inside $\mathcal{V}$.

To determine insideness, previous approaches either require watertight inputs , use error-prone voxel flood-filling (Mescheder et al., 2019) or use inaccurate visual hulls as a proxy (Park et al., 2019) (see inset where visual hill signing can be shown to "close off" internal structure. Virtox (left) under CC BY. ). Alternatively, Atzmon & Lipman (2020a;b) advocate for a loss function based on *unsigned* distances.

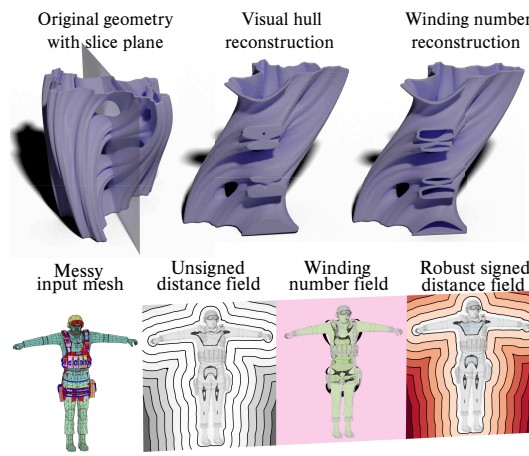

Original geometry with slice plane    Visual hull reconstruction    Winding number reconstruction

Messy input mesh    Unsigned distance field    Winding number field    Robust signed distance field

This introduces unnecessary initialization and convergence issues, that can be avoided if we assume that the input mesh intentionally oriented to enclose a solid region (as is the case for nearly all of Thingi10k), but may suffer from open boundaries, self-intersections, non-manifold elements, etc. Under these assumptions, the generalized winding number (Jacobson et al., 2013) computes correct insideness for solid meshes and gracefully degrades to a fractional value for messy input shapes (see inset). Using the tree-based fast winding numbers of Barill et al. (2018) and a bounding volume hierarchy for (unsigned) distances, we can construct our 1M-point sample set efficiently and optimize weights $\theta$ for even the most problematic meshes (see inset) in an average of 90 seconds per shape.

## 2.3 Efficient Visualization

Our weight-encoded neural implicit representation can be treated as its classical counterpart (SDF) and rendered efficiently using sphere-tracing (Hart, 1996). Sphere tracing is a common technique for rendering implicit fields where rays are initialized in the image plane and iteratively "marched" along by a step size equal to the signed distance function value at the current location. The ray is declared to have hit the surface when sufficiently close ($< \epsilon$). For more details, see Morgan McGuire's comprehensive notes at casual-effects.com.

We trivially adapt traditional sphere-tracing by initializing the starting position of each ray to be its first (if any) intersection with the similarity transformed unit sphere, since all weight-encoded neural implicits are normalized to lay within. As rays of the image will converge different times, we employ a dynamic batching method that composes batches of points for inference based on a mask buffer which tracks rays that have converged to the surface or reached the maximum number of steps. Local shading requires the surface normal at the hit point. For SDFs, the unit normal vector is immediately revealed as the spatial gradient (i.e., $\partial f_\theta / \partial \vec{x}$). This can be computed by finite differences or back propagation through the network.

## 3 Implementation and Results

We implement weight-encoded neural implicit networks in Tensorflow (Abadi et al. (2015)) with point sampling and mesh processing implemented in libigl (Jacobson et al. (2016)). We train our model for up to $10^2$ epochs and allow early stopping for quickly converging geometries. We use the ADAM optimizer (Kingma & Ba (2014)) with a fixed learning rate of $10^{-4}$. These settings generalized well across a wide range of geometries (see Figures 4 and 5).

## 3.1 Surface Visualization and CSG

We implement sphere-marching visualization and shading kernels in CUDA, using CUTLASS (Kerr et al. (2018)) linear algebra libraries for efficient matrix multiplication at inference-time.

We achieve an average display frame rate of *34 Hz* – for the large subset of the Thingi10k dataset we visualize – when rendering a single neural implicit at $512 \times 512$ resolution on an Nvidia P100 GPU. This a significant performance improvement over previous learnt implicit inference and display pipelines, attributed in large part to our compact representation. Liu et al. (2020) present a specialized renderer capable of a 1 Hz display rate, however at the price of many conservative optimizations: these include overstepping along all rays by a factor of 50%, increasing the convergence criteria (early stopping), and implementing a coarse-to-fine display strategy. While these additional optimizations could further improve our rendering speed (at the cost of reduced visual quality), we opt to rely on a simpler (and very efficient) standard sphere-marching SDF renderer.

Indeed, as our representation is a learnt representation of the SDF, we also inherit other important 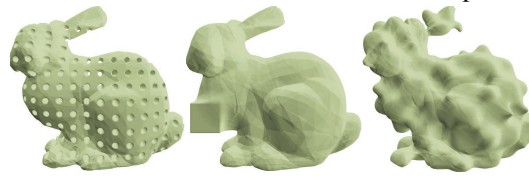 benefits of traditional implicit function representations. Weight-encoded neural implicits admit robust shape manipulation and modification using constructive solid geometry operations (CSG) – by directly modifying the inferred distance values (see inset and accompanying video). Weight-encoded neural implicits admit SIMD evaluation and, given their compactness, many neural implicits can be rendered in parallel at interactive rates on modern GPUs.

## 3.2 STABILITY AND SCALE

Training deep neural networks on large geometric datasets can be cumbersome and time consuming. For our weight-encoded neural implicit representation to be effective, we must be able to convert any 3D shape into its weight-encoded form in a reasonable amount of time. Due to our relatively simple base network architecture (8 layers of 32 neurons each) we find that we can overfit our model to any 3D shape **in 90 seconds**, on average. As this requires only 59 kB of memory, we can train many models/shapes concurrently on modern GPUs without approaching any practical memory limitations – this ease of training is uncommon to other learning-based shape representations. Converting the entirety of the 10,000 models in the Thingi10k dataset Zhou & Jacobson (2016) on an Nvidia Titan RTX only took 16 hours on a single GPU, or four hours on four Nvidia Titan RTX cards.

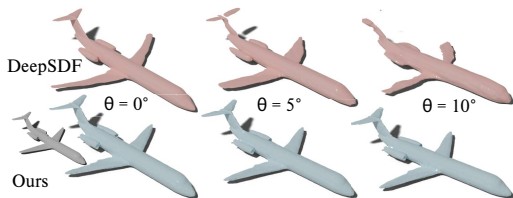

Figure 2: Unlike our representation, DeepSDF reconstruction quality degrades quickly for geometries not aligned to default, per-class orientations. See accompanying video for animation.

Converting the Thingi10k dataset from mesh format to weight-encoded neural implicit format reduces the overall storage from 38.85 GB to 590 MB – a 1:66 compression rate. While a DeepSDF network Park et al. (2019) trained on the same dataset could compress this dataset to an impressive 7 MB footprint, the latent-decoded geometries it produces are of comparably lower quality. This comparison is representative, as Thingi10k is a real-world mesh dataset of objects obtained "in the wild". The dataset neither contains geometries aligned to a common frame of reference nor comprises objects nearing no semblance of inter-class categorization. These two properties make it difficult for any latent-encoded neural implicit network to converge to a reasonable result during training.

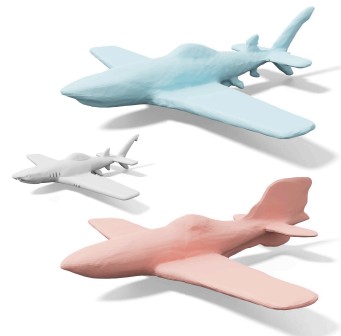

Figure 3: Latent-encoded SDFs (red) struggle to reconstruct "unique" features (grey, plane's tail) despite training on a single class of objects (planes). Our representation (blue) does not. gpvillamil under CC BY.

We further support these claims using two experiments. First, we attempt to train DeepSDF on the Thingi10k dataset, and second we experiment with DeepSDF's ability to reconstruct shapes with slight perturbations from the shapenet (Chang et al., 2015) common shape orientation. Here, DeepSDF does not converge on the 10,000 model Thingi10k dataset, producing incoherent reconstructions when exploring the latent space of shapes it has learned. Moreover, if we further limit DeepSDF to training with a single class of objects, it is not able to reconstruct features on the tails of the inter-class distribution (inset, right). Secondly, we evaluate DeepSDF's ability to reconstruct geometries not aligned to the common orientation. Here, we retain single-class DeepSDF training and reconstruct the same input shape at orientations differing from the default (Figure 2). This test validates latent-encoding's reliance on having consistently aligned datasets, immediately precluding their use with large, real-world datasets.

## 3.3 REPRESENTATION COMPACTNESS

All of the shapes in Figure 4 were rendered with weight-encoded neural implicits generated using our base network architecture, resulting in a total of 7553 weights for each shape's implicit function. At just 59 kB of memory we find that our lightweight representation can capture complex geometric topologies at high resolution compared to uniform signed distance grids or adaptively decimated meshes with similar memory footprints.

The comparisons in Figure 8 use geometry converted to a weight-encoded neural implicit in our base configuration, visualized next to the rendered result of a uniformly sampled SDF grid with $20^3$ samples as well as with the original mesh adaptively decimated (Garland & Heckbert, 1997) down to 7600 floats (i.e., vertex and face data). Compared to decimated meshes (our baseline non-uniform format), we observe that weight-encoded neural implicits have similar surface

quality but with smoother reconstructions due to the continuous (versus piecewise linear) nature of the implicit. Compared to SDFs stored on a grid (our baseline uniform format) we observe far better quality at equal memory. Furthermore, we notice that our approach better captures high frequency surface detail compared to both these representations, often producing results that more closely match the curvature of the original shape.

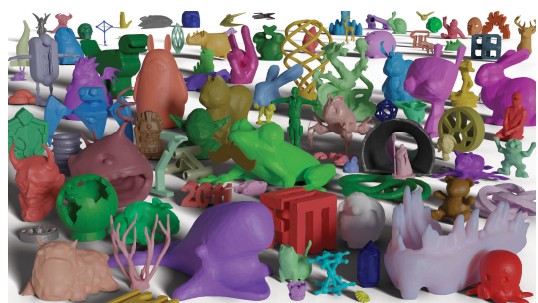

Figure 4: Thingi10k models compressed to **59kB**, reducing the dataset from 38.85 GB to 590 MB.

We measure our method's robustness by converting the Thingi10k ((Zhou & Jacobson, 2016)) dataset and measuring the average surface error $(1/N \sum_{i=1}^{N} |f_\theta(p_i)|)$ and training loss. We report mean training loss for errors between the true and predicted SDF values at points sampled using our importance metric (Section 2.1.2). This surface error is the sum of errors at points along the shape's 0-isocontour. These metrics measure both the error at the surface and within the shape's bounding volume. Errors within the bounding volume decrease rendering performance and/or lead to hole artifacts in the shape during visualization. Surface errors are more evident after meshing the implicit SDF using, i.e., marching cubes. We sample $10^5$ surface points when measuring surface error, and compute loss against a training set of $1M$ points. We visualize results on the entire Thingi10k dataset in Figure 5. We find that, at our base configuration, 93% of the $10^5$ Thingi10k shapes reach a surface error below 0.003, and no model exceeds 0.01 (worst case of 0.0097; see Fig. 6).

## 4 LIMITATIONS AND FUTURE WORK

Our default architecture will fail to satisfyingly approximate very topologically or geometrically complex shapes. While increasing the size of the network will generally alleviate this (see Figure 6), it would be interesting to consider cascading or adaptively sized networks. Our $L^1$ loss function encourages the network to match the *values* of a shape's signed distance field,

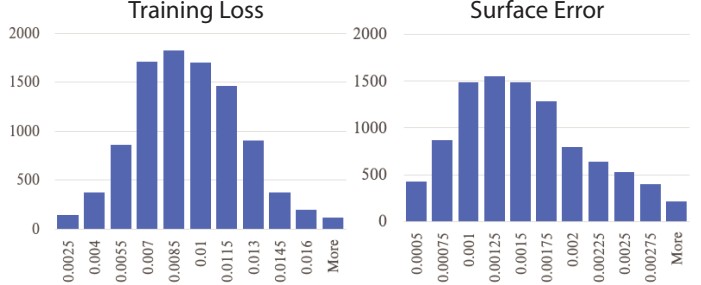

Figure 5: Loss and surface error distributions over the entirety of the Thingi10k dataset.

but not necessarily its derivatives (cf. Gropp et al. (2020); Sitzmann et al. (2020b)). True SDFs satisfy an Eikonal equation ($|\partial g/d\vec{x}| = 1$) and this property is sometimes important for downstream tasks. For future work, we would like to investigate whether Eikonal satisfaction can be ensured exactly by construction. With respect to single-shape accuracy, latent-encodings work well in specialized scenarios (e.g., large-networks trained on canonically aligned specialized classes). With respect to shape-space learning, latent-encodings lie in a simpler continuous space than weights, which suffer from transposition and reordering non-injectivities (i.e., multiple weight vectors represent the same implicit). Nevertheless, weight-encodings allow us to faithfully prepare large diverse datasets of 'real-world' shapes into a vectorizeable representation. We have shown this is simply *not* possible with existing latent-encodings. We include the full Thingi10k dataset converted to weight-encoded neural implicits vectors as a data release[2]. This vectorized data is ripe for meta-learning future work. Indeed, concurrent work is already exploring this direction Sitzmann et al. (2020a). We hope our consideration of weight-encoded neural implicits as a first-class shape representation encourages their use in computer graphics, geometry processing, machine learning, and beyond.

---

[2]https://github.com/u2ni/ICLR2021

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

## A  APPENDIX

### A.1  THE WEIGHT-ENCODED NEURAL IMPLICIT FILE FORMAT

Our compact weight-encoded neural implicit is designed to be effortlessly consumed and integrated into existing graphics and geometry processing pipelines. For each trained model, the chosen architecture and similarity transformaton matrix (since all geometries are normalized to the unit sphere) are written as the first bytes before encoding the learned weights $\theta$ into an HDF5 format file.

For a fixed architecture, the instructions to evaluate the estimated SDF is the same for any point *and* any shape. This SIMD property allows multiple geometries to be evaluated in parallel. The fixed storage profiles and memory layout of our learned implicit functions provide consistent query and rendering speeds. We store our model weights using the HDF5 format. This allows easy integration into Tensorflow (below) which can load our model natively. We additionally support the loading of arbitrary weight-encoded neural implicit through the "High Five" HDF5 C++ library (https://github.com/BlueBrain/HighFive) for rendering and meshing.

```
import tensorflow as tf
import numpy as np

# load model "key" dictating architecture. SIMD.
sdfModel = tf.keras.models.model_from_json(open('key.json'))

# load specific weight for Standford bunny geometry
sdfModel.load_weights('bunny.h5')

# generate 128x128x128 grid for SDF queries
K = np.linspace(-1.0,1.0,128)
grid = [[x,y,z] for x in K for y in K for z in K]

# infer SDF at each point
S = sdfModel.predict(grid)
```

## A.2  ERROR DRIVEN CONVERSION

We fix the architecture during the Thingi10k dataset conversion, resulting in a constant and compact memory footprint. If, however, maintaining a target surface reconstruction quality is of more importance to a fixed memory cost, we can instead shift to an error driven surface fitting approach (much like classical approaches (Ohtake et al., 2005)), scaling network architecture complexity based on the input geometry. As each generated weight-encoded neural implicit encodes its own architecture, such an approach results in smaller architectures for simpler geometries and larger ones for topologically-complex geometries. We visualize the effect of error driven optimization in Figure 6, where we perform a simple grid search until reaching a user-desired surface error threshold.

Based on our conversion of the Thingi10k dataset, we find that a majority of models are well represented using our base configuration (Fig. 5) – if desired, geometries that fall within the tails of the complexity distribution can be retrained with larger architectures, again until we reach a desired surface fidelity. This decision can be further informed by whether SIMD and fixed memory access patterns are beneficial to the underlying application.

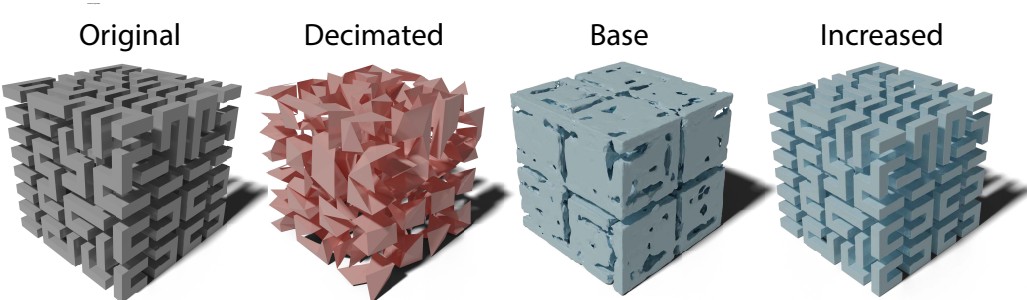

Figure 6: With only 7553 parameters, our *base* weight-encoded neural implicit format can lack the representative power to converge on highly complex geometries (similar to decimated mesh with same memory footprint). Increasing the network capacity to equal the memory impact of the original mesh results in near perfect reconstructions. tbuser (left) under CC BY.

## A.3  SIREN AND FOURIER FEATURES

In an effort to improve the reconstruction quality of our weight-encoded neural implicits we explored recent work focused on improving MLPs ability to represent high frequency signals. We experimented with three methods: namely we investigated using the SIREN (Sitzmann et al., 2020b) activations, positional encoding (Mildenhall et al., 2020), and Fourier features (Tancik et al., 2020). Each of these approaches have lead to impressive resuts for high-fidelity reconstructions of 3D surfaces mitagating the known problem that MLPs learn low frequency signals faster (Rahaman et al., 2019).

Mildenhall et al. (2020) define their positional encodings as,

$$\gamma(p) = (sin(2^0\pi p), cos(2^0\pi p), ..., sin(2^{L-1}\pi p), cos(2^{L-1}\pi p)) \tag{8}$$

where $\gamma$ is a mapping from $\mathbb{R}$ into the higher dimensional space $\mathbb{R}^{2L}$.

While Tancik et al. (2020) expands on this approach with random gaussian features yielding the mapping function,

$$\lambda(p) = (cos(2\pi\,\mathbf{B}p), sin(2\pi\mathbf{B}p)) \tag{9}$$

where each entry in $\mathbf{B} \in \mathbb{R}^{m \times d}$ is sampled from $\mathcal{N}(0, \sigma^2)$, and $\sigma$ is left as a hyperparameter specific to each problem.

We evaluate both of these approaches by mapping each axis (x,y,z) of our sampled points to the higher dimensional space. We find that when the network architecture is of sufficient width these mappings work exceptionally well. We evaluated using $\gamma$ with various $L$ configurations ranging from 4 to 10. Unfortunatly, we find that our light weight (and intentionally underparamerterized) architecture struggles to learn from the augmented input signal. We visualize the affect of positional encodings when $L = 10$ in Figure 7. Similarly, we see drastic degredation of quality when employing $\lambda$ for mapping to a default embedding size of 256 (not shown as we were unable to march). These approaches are clearly practical methods for reconstructing high-fidelity surfaces, but with our focus on minimizing the number of parameters the cost of mapping to a higher dimension input is too high.

Our experimental setup for evaluating Sitzmann et al. (2020b) periodic activation consisted of modifying an existing tensorflow (Abadi et al., 2015) implementation to accept our spatial queries as input and signed distances as target. We train the SIREN model to 200 epochs with a learning rate of $5e^{-5}$ and the same loss as our own configuration. Interestingly, we find that the SIREN model produces smoother approximations of the armadillo's surface (see Figure 7) but lacks fine detail. Once again, when increasing our model complexity to just 8 layers of 64 hidden units, we start to see the benefits of the periodic activation yielding much better approximations of the surface then our relu activation. For our base configuration of just 7553 parameters we choose to continue using RELU activation, but where high-fidelity weight-encoding neural implicits are required, SIREN should be employed.

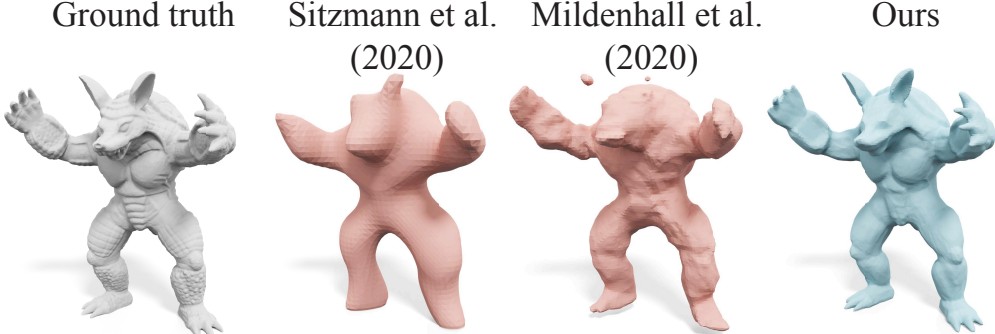

| Ground truth | Sitzmann et al. (2020) | Mildenhall et al. (2020) | Ours |

Figure 7: Results of using Mildenhall et al. (2020) and Sitzmann et al. (2020b)

## A.4 REPRESENTATION COMPACTNESS

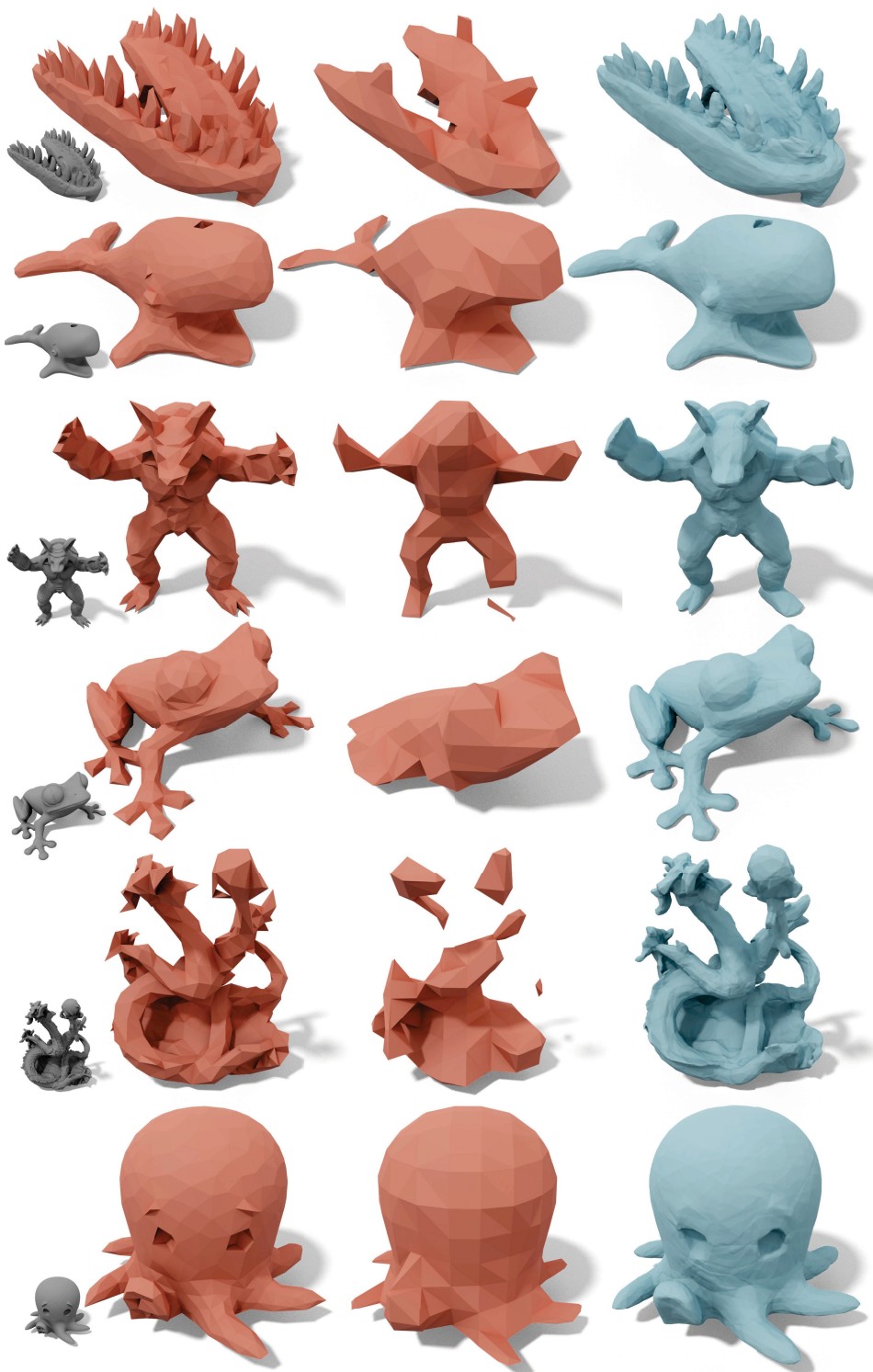

Figure 8: Our learnt weight-encoded neural implicit format (right) can be shown to better approximate the original surface (grey, inset) compared to adaptive decimation of the original triangle mesh Garland & Heckbert (1997) (left) and uniform signed distance grid (middle) with equal memory impact. gpvillamil (skull), Makerbot (whale), morenaP (frog), artec3d (dragon), JuliaTruchsess(octopus) under CC BY.

