# OpenReview forum: "On the Effectiveness of Weight-Encoded Neural Implicit 3D Shapes   "
_ICLR.cc/2021/Conference — Reject_

### Official Review · AnonReviewer3 · 2020-10-16
**Weight-encoded neural implicit representation for 3D shapes**

**Rating:** 8
**Confidence:** 4

**Review:**

The paper proposes a weight-encoded neural implicit representation for 3D shapes. The idea is to encode every shape in the network weights of its own designated small MLP network, instead of trying to learn a latent space of shapes. This leads to a really compact shape representation based on signed distance fields that could be interesting for many applications. The approach uses importance sampling to speed up training and robust losses.

The paper evaluates the representation in terms of visualization efficiently, stability, and the compactness of the representation on a large dataset of shapes. The approach outperforms latent encoded shape representations such as DeepSDF in terms of compression and accuracy.

I believe the idea of encoding shapes using signed distance fields and MLPs for compression is really exciting. Such implicit representations of shapes are becoming really popular at the moment and have huge potential in the future. In terms of novelty, I have seen many shape representation approaches based on MLPs lately, but none so far focussed on the compression aspect.

In summary, I believe that the compression aspect of this work is really interesting and will inspire follow up works along those lines. The method achieves impressive compression ratios and can decompress/render the shapes still at high speeds.

---

> ### Author Response · Authors · 2020-11-17
> **Reply to AnonReviewer3**
>
> We thank the reviewer for their comments and interest in our work. We too are excited about the potential future work on our neural implicit format and are pleased to release our code and datasets for the community to explore.

---

### Official Review · AnonReviewer2 · 2020-10-26
**some interesting idea, but meeds more work on presentation and evaluations**

**Rating:** 4
**Confidence:** 3

**Review:**

This paper discusses implicit SDF representations encoded by neural networks.

The paper is written partly as an opinion paper, partly as if it was an enormous contribution and completely new idea to overfit a network to a simple shape (thus not requiring a latent vector) compared to using a latent code to obtain a common latent space for a set of shapes. It is thus constantly written in opposition to Park et al. 2019, which I found useless and tiring to read, takes far too much space to my taste and I think is detrimental to the overall quality and interest. The idea of encoding an implicit function in the weights of a network without latent code is not new at all, and in particular common practice in all the NERF literature. The fact that it will lead to better reconstructions than learning a common latent space is obvious.

To me, the most interesting parts of the paper is the tools it proposes to deal with implicit SDFs, that are, independently of this paper, becoming more important in 3D vision: the use of generalized winding numbers to encode general meshes and efficient visualization tools. Admittedly, these are known in the graphics community, which may make the paper hard to publish in a top tier conference, but I believe modern tools compatible with recent DL libraries would be valuable and a paper introducing them could have a good impact. Similarly, a pipeline allowing CSG operations could have been presented and analyzed (further than a few qualitative examples)  The current version of the paper however provides too little details and evaluation on these and on the contrary spends a lot of time on discussion which I felt were not any real contribution.

In more details, I from my point of view
- sections discussing relation to deepSDF (1 and 2.1.1) could be drastically compressed (and section 1 could be rewritten completely)
- the paper is missing a related work
- the interest of 2.1.2 is very limited. Yes, sampling is equivalent to weighting, this is more or less importance sampling. Some papers prefer to define the sampling, this one prefers to define the weight. The results to support this strategy seem anecdotical and insufficient (one would have to carefully cross-validate the parameters in both to draw any conclusion) Moreover, it is far more costly since need to start by sampling a much larger set of points.
- I find 2.2 and 2.3 of potential interest, but they are not much developed nor evaluated
- 3.1 is missing the crucial evaluation of the rendering quality
- 3.2 focusses on comparison with deepSDF which I think does not bring much since it is a weak baseline/not designed for the same purpose + should be quantitative, not qualitative
- I did not find 3.3 to be very informative
-  the paper is missing clear evaluations and quantitative analysis (which is related to the fact its contribution is unclear)
- the conclusion (4) finally discusses the issue of normals, which I believe should have been addressed earlier



Some more comments:
- I think the "neural implicit" term introduced near the first inset is not well chosen (implicit is an adjective) but in any case it should not be used so much in the intro before its introduction
- the quality of the reconstructions is actually not great, this is not very visible in a printed version because all images are small, but zooming on the bunny for example actually shows it is quite coarse. This is an important practical problem, and one could imagine how to jointly work efficiently with representation of different quality (even if that sounds more like a graphics paper than an ICLR paper) Also, from the video it seems training is not super stable (which is surprising, maybe decreasing the LR after a number of epochs would lead to better results)
- the MLP encodes a linear by part function, the zero-level-set   (which does not change after the tanh) is thus a set of planes, resulting in a planar by part approximation of the shape, which is very visible in the qualitative results and should be discussed. (this also implies that the same quality for a similar budget can be obtained with a mesh)
- 90 seconds to obtain a representation is still very expensive. Indeed, if one were, as suggested in the paper, to use the representation to perform CSG, one would need to either save the full construction process or convert often into the implicit representation. It's also unclear how errors would accumulate when re-encoding several time a given shape.

---

> ### Author Response · Authors · 2020-11-17
> **Reply to AnonReviewer2 (1 of 2)**
>
> We thank the reviewer for their thorough and useful feedback addressing the main concerns below. We hope the reviewer will reconsider their assessment after considering our elaborations, below.
>
> ***“The paper is written partly as an opinion paper, partly as if it was an enormous contribution and completely new idea to overfit a network to a simple shape”
> “Sections discussing relation to deepSDF (1 and 2.1.1) could be drastically compressed (and section 1 could be rewritten completely)”
> “idea of encoding an implicit function in the weights of a network without latent code is not new at all”***
>
> We do not claim to have invented overfitting a neural network to a single shape. In section 1, we discuss the comparison to DeepSDF (Park et al. 2019), IM-NET(Zhang & Chen, 2020), and SAL (Atzmon & Lipman 2020) specifically to show that this is in fact not a new direction but, rather, an overlooked one. In each of these methods weight-encoded neural networks are used simply as a basic validation step, and little evaluation or emphasis is put into their potential value. We hope our paper serves to show that weight-encoded neural implicits can be both feasible and very useful.
>
> NeRF (Mildenhall et al. 2020) offers an incredibly powerful method of representing complex scenes as a volumetric scene for novel view synthesis. These methods provide impressive reconstructions but cannot be trivially integrated into the graphics pipeline, and would be completely infeasible to train on large datasets. In fact with NeRF’s 1-2 day training time per model it would take over 27 years to translate the Thingi10k (Zhou & Jacobson, 2016) dataset resulting in a (un)compressed dataset of 50 GB from the original 38 GB. This is in contrast to our weight-encoded neural implicit taking just 16 hours and reducing the dataset to just 590 MB.
>
> ***“3.2 focusses on comparison with deepSDF which I think does not bring much since it is a weak baseline/not designed for the same purpose + should be quantitative, not qualitative”***
>
> We respectfully disagree with the reviewer on this note. The rotation and class dependence of DeepSDF is a phenomenological problem. The qualitative comparison figures shows this much better than a quantitative experiment. We compare against DeepSDF to showcase the limitations of latent-encoded neural implicits in contrast to weight-encoded ones. In general, we show that for diverse datasets of geometries in arbitrary orientations and of non-related categories latent-encoded neural implicits simply cannot converge due to their specifically engineered focus on learning the “space of shapes”. Weight-encoded neural implicits overfit to a single shape and converges regardless of orientation or variety of the dataset. Figures 2 and 3 are designed to convey this.
>
>
> ***“the most interesting parts of the paper is the tools it proposes to deal with implicit SDFs”***
>
> Thank you for this comment. We feel that incorporating more robust geometry processing tools into the rapidly developing field of 3D geometric learning is crucial to ensure our methods can support real world data, not just beat benchmarks on curated datasets of watertight meshes (i.e., ShapeNetCore).
>
> ***“a pipeline allowing CSG operations could have been presented and analyzed”***
>
> Our CUDA renderer is simply based on traditional sphere tracing implementations. We chose to first write our renderer for arbitrary volumetric fields (from a uniform grid of SDFs) before modifying it to support queries against our weight-encoded neural implicits instead. The main point here is that our neural implicit format is so compact that it can easily be loaded into GPU memory as if it were a traditional implicit field. Where other methods are forced to make quality sacrificing simplifications (Liu et al. 2020), we find that our method is able to achieve 34 Hz rendering with no changes to the pipeline and with a relatively unoptimized implementation.
>
> ***“The paper is missing a related work [section]”***
>
> Due to the position of our paper offering a deep dive into an existing idea, we choose to combine our introduction and related work. If this concern rather refers to  us missing a specific reference, then please let us know and we can revise our introduction to include any additional related works.
>
> ***“sampling is equivalent to weighting, this is more or less importance sampling. Some papers prefer to define the sampling, this one prefers to define the weight… Moreover, it is far more costly since [we] need to start by sampling a much larger set of point”***
>
> Our point is not about preference. Ad hoc weighting or sampling schemes obscure or may not even correspond to an underlying integrated loss function. Unlike prior methods we make this connection clear by demonstrating how sampling and weighting follows from Monte Carlo approximation of a continuous integral. The cost of oversampling for our importance sampling is negligible compared to the other aspects of training.

---

> ### Author Response · Authors · 2020-11-17
> **Reply to AnonReviwer2 (2 of 2)**
>
> ***“I find 2.2 and 2.3 of potential interest, but they are not much developed nor evaluated”***
>
> We thank the reviewer for their interest in our robust mesh tools for geometries in the wild and our modified ray marcher for neural implicits. We choose to keep both these sections brief since, indeed, these methods are well known in the graphics community. For a more detailed look into winding numbers see: “Robust inside-outside segmentation using generalized winding numbers.” by Jacobson et al. While for a more thorough understanding of our rendering pipeline see McGuire’s notes: https://casual-effects.com/research/McGuire2019ProcGen/McGuire2019ProcGen.pdf.
>
> ***“3.1 is missing the crucial evaluation of the rendering quality”***
>
> Renders are generated using a classical volumetric rendering technique known as ray marching. At a high level, we can render a single pixel by initializing a ray at the eye and through the image plane and stepping along the ray a distance equal to the signed distance at the current position. This process iterates until the current position on ray is either “close enough” to the surface or has taken too many steps. In our case the signed distance at current position is obtained by inferring against our weight-encoded neural implicit. The quality of the render is directly correlated to the training loss achieved in training. We would like to mention however, that we are excited by the idea of integrating a differentiable ray marcher into our training process to further improve the converged result specifically for rendering applications.
>
> ***“From the video it seems training is not super stable”***
>
> Thanks for the comment. This is purely an artifact of the rate at which we rendered the frames (i.e., not every frame) and not an indicator of the training stability. We will update the video with a more frequent render during the training process.
>
> ***“quality of the reconstructions is actually not great”***
>
> We include the full Thingi10k dataset of 10,000 geometries to be transparent in the reconstruction quality of our compact neural implicit format. In Figure 8, we show how our representation compares to both adaptively decimated triangle mesh and uniform signed distance grids of the same memory impact (59 kB). Compared to adaptively decimated mesh, we observe that weight-encoded neural implicits have similar surface quality but smoother reconstructions due to the continuous (versus piecewise linear) nature of the learnt implicit. While compared to SDFs on a uniform grid we observe far better reconstructions.
>
> It is important to note that higher quality reconstructions can always be achieved by increasing the network capacity (as shown in Figure 6) at the cost of compactness and inference speed (supported by Figure 1).
>
> ***“90 seconds to obtain a representation is still very expensive.”***
>
> It is important to note here the immediate benefit of weight-encoded neural implicits overfitted to a single shape at a time. In DeepSDF the latent-encoded neural implicit is trained to 1000 epochs on 8 GPUs taking approximately 8 hours. Then each new shape latent vector optimization takes on average 8s. Based on our translation of the Thingi10k dataset outlined in section 3.2 taking 16 hours on a single GPU (with 16 neural implicits trained in parallel), in 8 hours with 8 GPUs we could convert over 40000 geometries to the weight-encoded neural implicit format compared to DeepSDFs training set of 4858 (we assume the reported 8 hour training time is on Table objects, the largest category in shapenet).
>
> Additionally, when compared to NERF (a network for high resolution reconstructions, not compactness) we find our 90s (single model single GPU) conversion rate producing a 59 kB weight set to be favorable to the 1-2 days training procedure yielding a 5 MB encoding. As mentioned earlier, NERF would take over 27 years to translate the entire Thingi10k dataset.
>
> ***“if one were… to use the representation to perform CSG one would need to either save the full construction process or convert often into the implicit representation…”It's also unclear how errors would accumulate when re-encoding several time a given shape.”***
>
> One of the beneficial properties with implicits is that we don’t need to re-encode the entire CSG modified implicit after every operation. The entire CSG tree can be evaluated (and rendered) using leaf-node implicits. We introduce the CSG operations in section 3.1 to showcase that neural implicits, like traditional implicits, admit robust shape manipulation and modification using standard CSG operations. Re encoding the result of a large CSG tree as a Neural Implicit is an interesting problem that we leave for future work.

---

### Official Review · AnonReviewer1 · 2020-10-30
**Enjoyable read**

**Rating:** 7
**Confidence:** 5

**Review:**


This paper presents a simple (and convincing) idea that neural networks can be employed to "memorize" the surface geometry of 3D objects. Such memorized networks -- dubbed neural implicits -- exhibit significantly lower memory requirements, while allowing for better surface approximation compared to alternative prior approaches.

## Strengths

**S1** This paper pursues a very timely research direction: several interesting approaches over the past year [A, B, C] (to list a few) have been proposed that leverage the overparameterized nature of neural nets to "memorize" the surface geometry of objects. While prior approaches require the neural network to learn a "distribution" over multiple shapes of a category and "generalize" to unseen shapes at test time, this paper poses a different question: can neural nets learn to reproduce one specific shape (by overfitting). This essentially turns a bug (i.e., overfitting) into a feature (i.e., neural implicits), which is a neat idea!

**S2** The paper is extremely well-written, and a thoroughly enjoyable read. Most design choices are well-motivated, and where applicable, empirically/qualitatively justified.

**S3** I find the discussion about various sampling strategies insightful. This section (2.1.2) makes apparent several intricacies that usually fly under the radar. Although I do have a few reservations (see discussion under "Weaknesses"), I think the paper does a good job of explaining how sampling strategies such as [A] result in unintended bias. It is worth noting that there isn't consensus in the implicit-representation-learning community about what sampling strategies work better than the others (eg. [A] and [B] present conflicting accounts on this matter).

**S4** In general, I find the experiment setup convincing. The code and/or data release might see a lot of community interest in following up on this work.

## Weaknesses

I do think this work can benefit from a number of additional discussions and/or explorations, which I list below.

**W1** It appears that the focus of this work is in reconstructing the geometry of the surface. Often 3D meshes have other important attributes that are essential for downstream tasks (face normals, color and/or texture information, material properties). This feels like an important limitation that could benefit from discussion. (Do we expect learning textures, normals, etc. to be "trivial"? In which case it'd greatly strengthen the paper. If not, explicitly highlighting this could result in interesting follow-up work).

**W2** While the paper paints the rosy side of neural implicits, it might benefit from quantifying the "ineffectiveness of weight-encoded neural implicit 3D shapes". How well do surface continuity properties hold (C0, C1, and higher) hold? What can possibly be done to mitigate such ill-effects (if any)? While Sec. 4 talks about these questions being interesting future directions, it might add value to the current submission to quantify the gap (eg. Eikonal constraint violation, watertight-ness or manifold-ness violations) to be comprehensive.

**W3** Traditionally, multiple representations have flourished in the computer graphics community, primarily because each of them possesses a complementary set of strengths and weaknesses. I'd assume that to hold true for neural implicits too. For instance, rendering neural implicits, computing higher order information (curvature? derivatives?) and above all, interactive editing applications might be more inefficient with neural implicits?

**W4** I feel the discussion about sampling strategies could greatly benefit from an accompanying empirical analysis. While most arguments in this section (2.1.2) appear intuitive, they're often hand-wavy.

**W5** Since one of the primary goals of neural implicits is to reduce memory requirements for storing and processing polygon meshes, I hoped to see more discussion on "classical" data compression techniques used in the computer graphics community. A seminal reference is [D].

**W6** Discussion of a few other closely related papers, such as [E, F] can be added.

## Minor remarks

**M1** The following remarks have had zero-impact on my overall score, and as such, I do not expect the authors to respond on these issues.

**M2** The abstract contains a few (minor) grammatical errors that cound benefit by a proofread.

**M3** A concurrent recent work [G] seems to follow a fairly similar idea, and it would be nice to discuss and contrast the current submission wrt this (particularly for the camera-ready version).


References
==========

[A] Mescheder, Lars, et al. "Occupancy networks: Learning 3d reconstruction in function space." Proceedings of the IEEE Conference on Computer Vision and Pattern Recognition. 2019.

[B] Park, Jeong Joon, et al. "Deepsdf: Learning continuous signed distance functions for shape representation." Proceedings of the IEEE Conference on Computer Vision and Pattern Recognition. 2019.

[C] Sitzmann, Vincent, et al. "Implicit neural representations with periodic activation functions." arXiv preprint arXiv:2006.09661 (2020).

[D] Pierre Alliez and Craig Gotsman. Recent Advances in Compression of 3D Meshes. Advances in multiresolution for geometric modelling, 2005.

[E] Hanocka, Rana, et al. "Point2Mesh: A Self-Prior for Deformable Meshes." arXiv preprint arXiv:2005.11084 (2020).

[F] Fourier Features Let Networks Learn High Frequency Functions in Low Dimensional Domains. Matthew Tancik*, Pratul Srinivasan*, Ben Mildenhall*, Sara Fridovich-Keil, Nithin Raghavan, Utkarsh Singhal, Ravi Ramamoorthi, Jonathan T. Barron, Ren Ng.

[G] Overfit Neural Networks as a Compact Shape Representation. arXiv. 2020.

---

> ### Author Response · Authors · 2020-11-17
> **Reply to AnonReviewer1**
>
> We thank the reviewer for their thorough and useful feedback and address their comments below.
>
> ***“Often 3D meshes have other important attributes that are essential for downstream tasks… Do we expect learning textures, normals, etc. to be trivial?”***
>
> While the focus of our work is purely on geometric representation, we do expect that learning colors or high frequency normal corrections is an incremental extension.
>
> ***“How well do surface continuity properties hold (C0, C1, and higher) hold? What can possibly be done to mitigate such ill-effects (if any)?”***
>
> The surface of the neural implicit is the level set of a continuous function and thus is always at least C0. Since every output will thus be watertight and manifold, we measure reconstruction quality by surface  error (quantified in Figure 5). Section 3.3 applies this surface error comparison on the Thingi10k (Zhou & Jacobson (2016)) dataset, where all models visualized in Figure 4 have a surface error below 0.003. We do note in Figure 6 however that failure cases can arise in practice when attempting to encode highly complex geometries into our compact representation. We also show (in Figure 6) that increasing the network capacity leads to better reconstructions. In our implementation we simply flag the user when surface error is below a certain threshold. We leave exploration of methods to guarantee satisfaction of the Eikonal equation as future work.
>
> ***“Traditionally, multiple representations have flourished in the computer graphics community, primarily because each of them possesses a complementary set of strengths and weaknesses. I'd assume that to hold true for neural implicits too.”***
>
> In this context we treat neural implicits to be a new improvement on classical implicit functions. Signed distance fields are a powerful implicit representation for modelling solids. In general, implicit surfaces have a number of advantages over explicit surfaces (mesh based)  during modelling such as: infinite resolution, solid geometry operations, domain repetition, and smooth blending. eural implicits inherit these advantages. One obvious drawback of all implicits (including neural implicits) is that many existing geometry processing techniques rely on a mesh format. An interesting future direction we would like to (or promote others to) explore is the application of Rohan Sawhney and Keenan Crane’s "Monte Carlo Geometry Processing" (Sawhney & Crane (2020)) to neural implicits.
>
> ***“I feel the discussion about sampling strategies could greatly benefit from an accompanying empirical analysis.”***
>
> Ad hoc weighting or sampling schemes obscure or may not even correspond to an underlying integrated loss function. Unlike prior methods we make this connection clear by demonstrating how sampling and weighting follows from Monte Carlo approximation of a continuous integral. We show how this approach enables complete control on the sampling bias at sampling time. Empirical analysis is limited as the comparison to uniform and surface sampling has been made many times before. Our sampling approach never harms the training process compared to surface sampling, while providing complete control over sampling bias. We will add a more thorough analysis between uniform sampling, surface sampling, and importance sampling to the appendix.
>
> ***“I hoped to see more discussion on "classical" data compression techniques used in the computer graphics community.”***
>
> We compare our method against adaptive decimation as our baseline. We choose to compare exclusively against decimated meshes and SDFs on grids as both of these representations can be immediately usable in an existing graphics pipeline. This is the same reason that we did not apply further compression algorithms on our learnt weight set, as our goal is to produce a compact and efficient representation that we can easily incorporate into the graphics pipeline.
>
> ***“Discussion of a few other closely related papers, such as [E, F] can be added.”***
>
> Thank you for the references; both of these papers are of interest to us. We discuss [F] (fourier features) in the appendix A.3, where we find that fourier features perform exceptionally well in improving high frequency reconstruction resulting in better quality edges and fine details, however -- after experimentation -- we note that these benefits require much larger networks than ours. We were excited to test both SIREN (Sitzmann el al. 2020) and Fourier Features and disappointed we could not yet obtain strong results in our regime.
>
> Although not immediately related, Hanocka et al.'s work on learning a self-prior on a single shape for point cloud reconstruction is an interesting work also showing the potential benefits of training neural nets on single geometries, and our manuscript will include additional discussing surrounding this work.

---

### Decision · Program_Chairs · 2021-01-07
**Final Decision**

**Decision:**

Reject

**Comment:**

The paper proposes how weight-encoded neural implicit can be strong 3D shape representations. A neural network is trained such that it overfits over a single shape, and the weights of such network is a great representation for the 3D shape. Results are shown on signed distance field (SDF) generation from meshes.

Strengths:
- an interesting idea for generating compact representations of 3D shapes
- Will further foster several conversations within the deep learning community

Weaknesses:
- Very limited evaluation to support the authors  claims, particularly against other traditional learnable 3D representations